# Phase-resolved measurement and control of ultrafast dynamics in terahertz electronic oscillators

Takashi Arikawa ⬤ [1,2,6] ✉, Jaeyong Kim[3,7], Toshikazu Mukai[3], Naoki Nishigami[4], Masayuki Fujita ⬤ [4], Tadao Nagatsuma ⬤ [4] & Koichiro Tanaka ⬤ [1,5]

As a key component for next-generation wireless communications (6 G and beyond), terahertz (THz) electronic oscillators are being actively developed. Precise and dynamic phase control of ultrafast THz waveforms is essential for high-speed beam steering and high-capacity data transmission. However, measurement and control of such ultrafast dynamic process is beyond the scope of electronics due to the limited bandwidth of the electronic equipment. Here we surpass this limit by applying photonic technology. Using a femtosecond laser, we generate offset-free THz pulses to phase-lock the electronic oscillators based on resonant tunneling diode. This enables us to perform phase-resolved measurement of the emitted THz electric field waveform in time-domain with sub-cycle time resolution. Ultrafast dynamic response such as anti-phase locking behaviour is observed, which is distinct from in-phase stimulated emission observed in laser oscillators. We also show that the dynamics follows the universal synchronization theory for limit cycle oscillators. This provides a basic guideline for dynamic phase control of THz electronic oscillators, enabling many key performance indicators to be achieved in the new era of 6 G and beyond.

Terahertz (THz) electronics is a rapidly evolving research field[1–6]. THz wave emitters based on compact electronic oscillators are suitable for real-world applications such as THz imaging[7], Radar[8], and wireless communications towards 6 G and beyond[4–6]. Several types of THz oscillators are developed using diodes[9–12] and transistors[13–15]. They use negative differential conductance (NDC) as an alternating-current gain to convert direct-current power to continuous wave (CW) THz radiation. The modern microfabrication technology was very successful in reducing the inductance ($L$) and capacitance ($C$) of the device, leading to the ultrahigh operating frequency ($\omega_0 = 1/\sqrt{LC}$) of THz. The highest fundamental oscillation approaching 2 THz is achieved[16] by resonant tunneling diode (RTD) oscillator[17–22]. However, the characterization of such ultrafast oscillation is extremely challenging. For a clear example,

THz frequency far exceeds the bandwidth of the state-of-the-art oscilloscopes, hindering the direct monitoring of the oscillation waveform[23].

At the same time, however, ultrafast time-domain measurement of the THz waveform is now routinely performed in the field of THz photonics. Instead of the electronic means, this technique is based on optical sampling such as electro-optic sampling[24] using femtosecond lasers and known as THz time-domain spectroscopy[25–28] (THz-TDS). In THz-TDS, a time resolution only limited by the pulse width of the femtosecond laser is achieved by utilizing phase-stable periodic nature of the signal waveform. Although THz-TDS is an ideal tool for characterizing THz electronic oscillators, no successful measurement has been reported so far due to their random phase fluctuation.

[1]Graduate School of Science, Kyoto University, Kyoto, Japan. [2]PRESTO, Japan Science and Technology Agency (JST), Saitama, Japan. [3]ROHM Co., Ltd., Kyoto, Japan. [4]Graduate School of Engineering Science, Osaka University, Toyonaka, Japan. [5]Institute for Integrated Cell-Material Sciences (iCeMS), Kyoto University, Kyoto, Japan. [6]Present address: Graduate School of Engineering, University of Hyogo, Himeji, Japan. [7]Present address: Qualitas semiconductor, co, ltd., Seongnam, Gyeonggi-Do, Republic of Korea. ✉e-mail: arikawa@eng.u-hyogo.ac.jp

In this paper, we show THz-TDS technique can be used to phase-lock electronic oscillators based on RTD by injection locking and perform optical sampling of the emitted THz waveform with a time resolution unachievable with electronic means. The phase-resolved measurement revealed the anti-phase locking nature, which is reproduced by the general synchronization theory for the limit cycle (Van der Pol) oscillators. We also demonstrated oscillation phase control through the injection signal waveform. Our findings will provide the foundation for phase control technology of THz electronic oscillators.

## Results

### Principle of optical sampling

To perform optical sampling like THz-TDS, it is essential that the same signal waveform is repetitively generated. In THz-TDS, this condition is satisfied because carrier-envelope phase stable THz pulses are generated by a mode-locked femtosecond laser[29,30]. In the frequency domain, this is equivalent to the offset-free comb spectrum at $nf_{rep}$ ($n$: integer, $f_{rep}$: laser repetition rate)[31]. In this case, THz wave with a period of $T_{THz} = (nf_{rep})^{-1}$ and sampling pulses derived from the same mode-locked laser with a repetition interval of $T_{rep} = (f_{rep})^{-1}$ are phase-locked, i.e., $T_{rep} = nT_{THz}$. Therefore, all the sampling pulses measure the same electric field value at a fixed phase (black solid circles on red solid waveforms in Fig. 1a) and this allows us to accumulate a huge number of tiny signals to sufficiently suppress the noise. By changing the relative timing between the THz and sampling pulse, we can reconstruct the whole waveform of the THz radiation.

Similarly, time-domain sampling of the CW THz wave from electronic oscillators is possible if the oscillation frequency matches $nf_{rep}$. However, this is generally unrealistic due to the phase fluctuation. In the case of RTD oscillators, the typical phase coherence time is less than one microsecond[32] (spectral linewidth on the order of a few MHz). In this situation, sampling pulses detect random electric field values every one microsecond (schematically shown by open circles on dashed red waveforms in Fig. 1a) and the required accumulation process produces zero signal. Therefore, phase locking is a key to realize optical sampling of electronic oscillators.

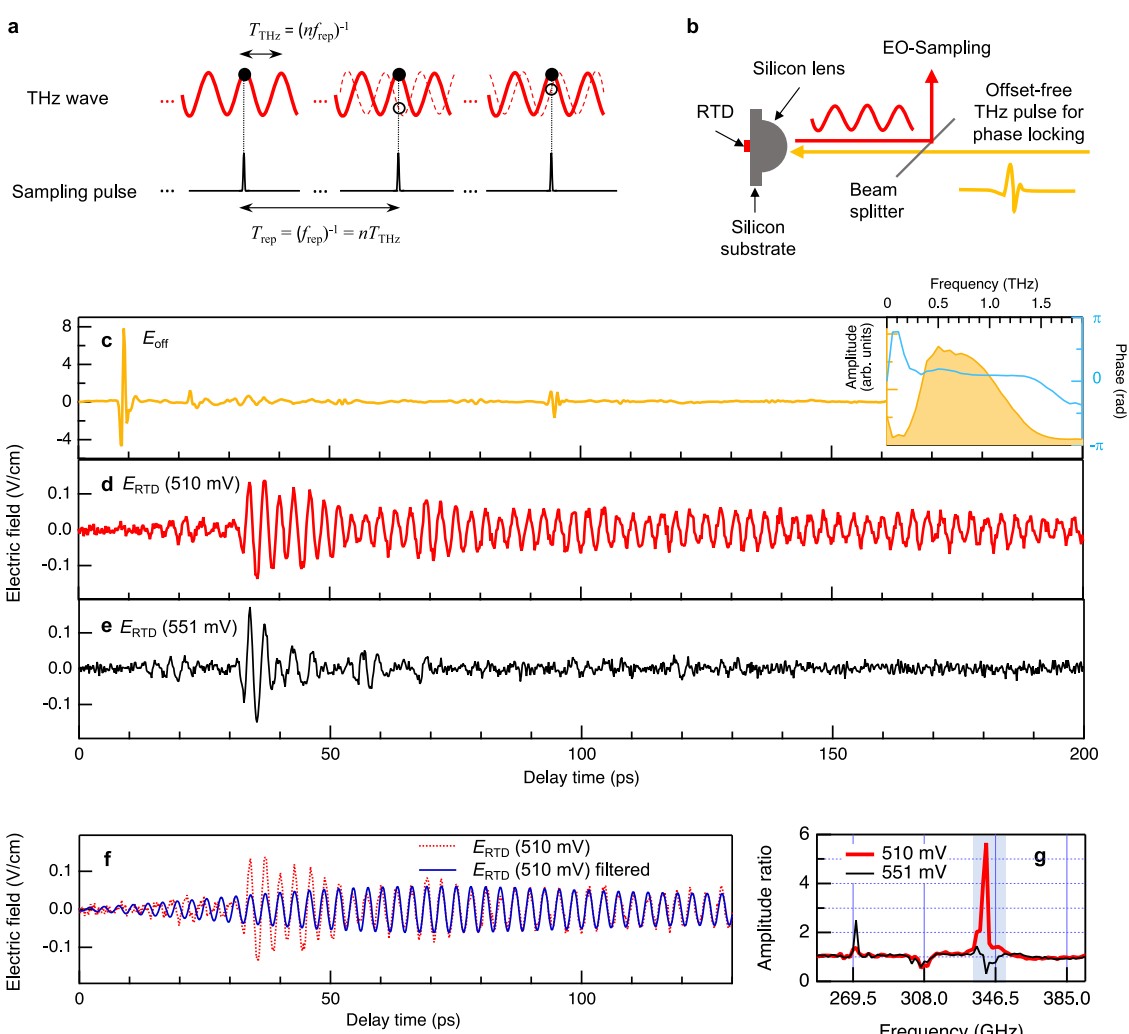

**Fig. 1 | Phase-resolved time-domain sampling of RTD THz oscillator. a** Principle of optical sampling. THz wave (solid red curve) and sampling pulses are phase-locked. Dashed red curve represents the THz wave from free-running electronic oscillators with phase noise. **b** Phase-locking of RTD by offset-free THz pulse injection. **c** Waveform of the injection THz pulse ($E_{off}$). The inset shows the amplitude (orange) and phase (blue) in the frequency-domain calculated from $E_{off}$ in the first 18 ps. The time origin was taken at 9 ps (positive peak of the main pulse) for the phase determination based on cosine function. Small-amplitude pulses at 22 ps and 94 ps are due to internal reflections inside the silicon substrate and detection crystal, respectively. **d** Waveform of the difference signal ($E_{RTD}$) at 510 mV. Injection THz pulse is nearly perfectly canceled by the double modulation technique. **e** Waveform of the difference signal at 551 mV which is just outside of the oscillation region. **f** Waveform of the RTD emission component (blue curve) at 510 mV obtained by frequency filtering. **g** Fourier amplitude ratio between bias voltage on and off (red: 510 mV, black: 551 mV). The vertical grid lines are drawn at the integer multiples of 38.5 GHz. The frequency resolution is 1.67 GHz. The amplitude ratio is almost unity in other frequency region. Only the spectral components in the blue shaded region are used to calculate the blue curve in d.

Recent research revealed that THz RTD oscillators can be phase-locked by injecting external THz wave with the amplitude as small as $10^{-4}$ of the RTD oscillator itself [32]. For typical RTD oscillators with 10 μW output power, this corresponds to the electric field amplitude on the order of μV/cm, which is achievable by THz pulses used in THz-TDS. In this study, we used offset-free THz pulses with a repetition rate of $f_{rep}$ for injection locking (Fig. 1b). This allows us to lock the oscillation frequency and phase of the RTD oscillator to $nf_{rep}$ and satisfy the requirement for the optical sampling.

## Time-domain measurement

We used an RTD oscillator made of AlAs/GaInAs double barrier structure [33] (see Methods and Supplementary Fig. S1 for detail). The size of the RTD chip is very tiny, comparable to a laser diode. The experimental setup is a standard THz-TDS system in reflection geometry (see Methods and Supplementary Fig. S2 for details). We injected THz pulses to the biased RTD and measured reflected THz wave ($E_{on}$) using electro-optic sampling technique. If the RTD oscillation is phase-locked by the THz pulse, we should be able to observe the electric field emitted by the RTD in addition to the reflected injection THz pulse. By subtracting the waveform of the injection THz pulse which can be measured without the bias voltage ($E_{off}$), we extracted the emission from the RTD ($E_{RTD} = E_{on} - E_{off}$). To avoid long-term fluctuation, we employed double modulation technique and obtained $E_{off}$ and $E_{RTD}$ simultaneously in a single delay stage scan rather than performing two scans for $E_{on}$ and $E_{off}$ (see Methods).

Figures 1c and 1d show the electric field waveforms of the injection THz pulse ($E_{off}$) and the difference signal ($E_{RTD}$), respectively. The bias voltage was set within the oscillation region (510 mV). Before the injection THz pulse comes at 9 ps, $E_{RTD}$ only shows noisy signal although the RTD is emitting THz wave. This is the consequence of the phase fluctuation described above. After the THz pulse injection, the difference signal starts to show sinusoidal oscillation with a frequency of 0.340 THz. This agrees with the free-running frequency and suggests the successful phase-locking and phase-resolved detection of the RTD emission in time domain. However, a similar waveform can also arise due to absorption. To ascertain the cause of the signal, we calculated the Fourier transform ($\tilde{E}_{on}$ and $\tilde{E}_{off}$) and checked the amplitude ratio ($|\tilde{E}_{on}|/|\tilde{E}_{off}|$). As shown by the red curve in Fig. 1g, a sharp peak above one at 0.340 THz is seen, indicating the presence of THz wave emitted from the RTD in addition to the incident THz wave. This allows us to eliminate the possibility of absorption as the cause for the oscillation observed in Fig. 1d.

Figure 1e shows the difference signal measured when the RTD was biased just outside of the oscillation region (551 mV). Even though the RTD was not oscillating, we still observed a finite signal. This is because the reflectivity of the RTD slightly changes by the bias voltage most likely due to the rectification by the RTD as a THz detector [34,35]. The sudden increase of the signal around 35 ps is explained by the internal reflection inside the device (26 ps round trip time. See "Internal reflection inside the RTD device" section in Supplementary information for details). A similar increase at 35 ps in Fig. 1d suggests that the signal due to the RTD reflectivity change is superimposed on the RTD emission. This point can be confirmed by comparing the amplitude ratio for both voltages (red and black traces in Fig. 1g); except for the RTD emission peak, they have small peaks and dips in common. These structures are almost equally spaced by 38.5 GHz (as indicated by the vertical grids), which is consistent with the existence of the multiple internal reflection of a round-trip time of 26 ps ($=38.5\,GHz^{-1}$).

## Injection locking dynamics

By eliminating the signal due to the RTD reflectivity change, we can extract the RTD emission component from the difference signal. The narrow spectrum of the RTD emission allows us to use a simple frequency filtering to suppress the signal due to the reflectivity change.

We numerically filtered out the frequency component outside of the blue shaded region in Fig. 1g and extracted the RTD emission component (blue trace in Fig. 1f). Although a small dip around 346.5 GHz is unavoidable, the large signal at 35 ps disappears and smooth buildup of the phase-locked component in about 50 ps is seen. For the decay dynamics, we don't have to rely on the numerical data processing because the signal due to the RTD reflectivity change becomes negligible after 80 ps (Fig. 1e). After 80 ps, the RTD emission amplitude gradually decreases and becomes almost constant after around 250 ps (for full waveform up to 600 ps, see Fig. 2a).

## Anti-phase locking

The electric field measurement in time domain allows us to determine the phase of the RTD oscillation. We fitted the RTD emission after 80 ps by a decaying cosine function with its time origin at 9 ps and obtained the initial phase of 1.10π. The phase of the injection signal at 0.340 THz is also obtained (0.11π) from the Fourier transform using cosine functions starting from 9 ps (inset of Fig. 1c). We found that the phase difference is 0.99π, which means that the RTD is anti-phase locked to the injection signal. These dynamical features will be explained by a general synchronization theory for limit cycle oscillators.

## Bias voltage dependence

We measured differential signals at different bias voltages to see how the RTD emission changes. Typical waveforms are shown in Fig. 2a (see Supplementary Fig. S3 for the complete data set). Similar dynamics as discussed above is observed for all bias voltages, but with different oscillation frequencies and amplitudes. Figure 2b shows the power spectrum ($|\tilde{E}_{RTD}|^2$) calculated from the time-domain data after 80 ps. The oscillation frequency determined from the peak position changes with the bias voltage as shown by the red circles in Fig. 2c, and follows the free-running oscillation frequency (blue trace). The intensity of the RTD emission component was evaluated from the peak area of the power spectrum. As plotted in Fig. 2d by the red circles, the overall trend is similar to the output intensity in the free-running state, measured by a conventional square law detector (blue), especially above 480 mV. The fluctuation below 480 mV is probably due to the interference effect that increases or decreases the signal as the oscillation frequency changes with the bias voltage. Finally, we determined the phase of the RTD oscillation at each bias voltage by applying the same fitting procedure described in the previous subsection. The phase of the injection THz wave at the RTD oscillation frequency can be obtained from the inset of Fig. 1c. By subtracting these two, the bias voltage dependence of the phase shift was determined, and the results are summarized in Fig. 2e. Although it shows some deviation, again below 480 mV, the data points are concentrated around π (i.e., anti-phase locking) regardless of the bias voltage.

## Effect of phase fluctuation

Although the electric field amplitude is stable up to 600 ps (Fig. 2a), it actually decays and becomes almost zero before 12.3 ns when the next injection THz pulse comes (at 81 MHz repetition rate). This is evident from the noise signal before the THz pulse injection shown in Fig. 1d. In this experiment, the differential signal decays due to the phase fluctuation. Therefore, the experimental results mean that the phase coherence time is shorter than 12.3 ns, which corresponds to the linewidth broader than 81 MHz. This is much broader than the typical linewidth of a few MHz [32], but can happen due to the noise from the power supply (in the current experiment, a function generator for double modulation measurement).

In some cases, however, we observed differential signals with longer coherence times than 12.3 ns. Figure 3 shows one example observed at 485 mV. We can see oscillation signal before the THz pulse injection, which is actually the RTD emission signal phase-locked by

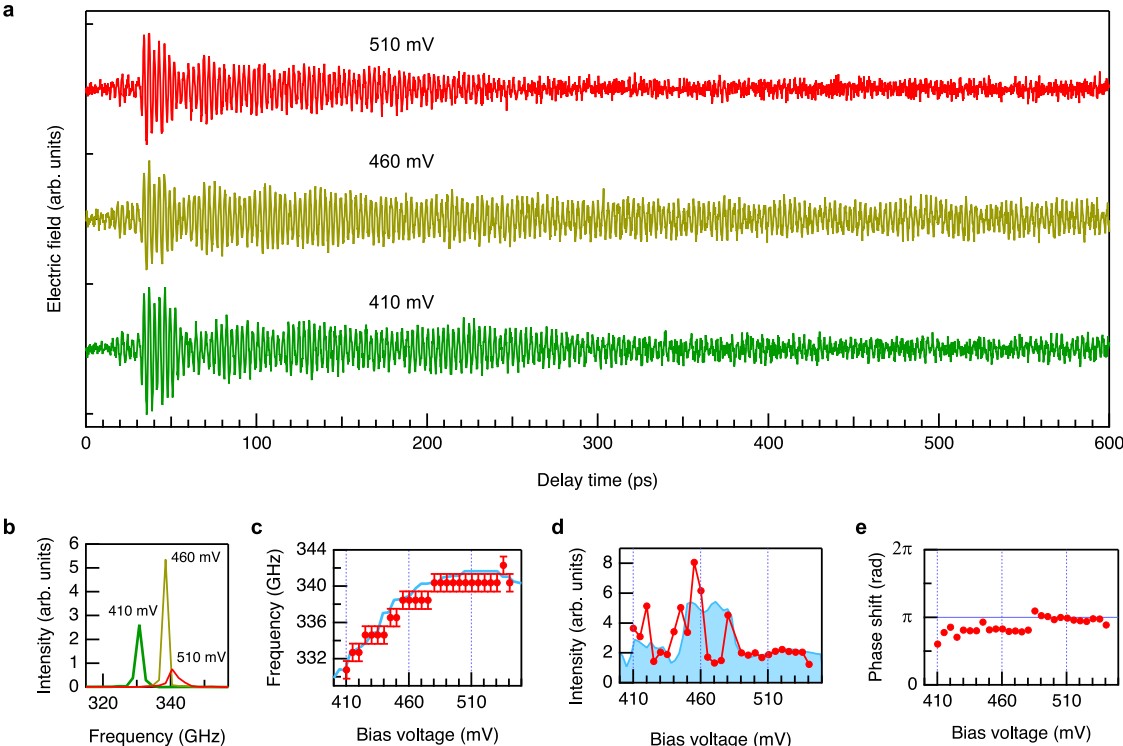

**Fig. 2 | Bias voltage dependence. a** Typical waveforms and (**b**) power spectra of the differential signals at three different bias voltages. To exclude the effect of the RTD reflectivity change, we used time-domain data after 80 ps to obtain the power spectra. (**c**) Oscillation frequencies determined from the peak position of the power spectra (red circles). The error bars represent frequency resolution (1.92 GHz). Blue curve represents the oscillation frequency in the free-running state determined by conventional heterodyne down-conversion method[52]. **d** Output intensity determined from the peak area of the power spectrum (red circles). Blue curve is the output intensity in the free-running state measured with a square law detector (Fermi-level managed barrier diode[53]). The data point at 485 mV is omitted due to the different situation of the phase-locking, as shown in Fig. 3. **e** Phase shift determined from the oscillation phase of the time-domain waveform at each bias voltage.

the previous injection THz pulses. This linewidth narrowing can be attributed to the effect of the cavity[32] formed by the RTD and optical components such as electro-optic crystal or photo-conductive antenna (see Supplementary Fig. S2a). This happens only when the RTD oscillation frequency coincides with one of the longitudinal modes of the cavity.

## Analysis based on Van der Pol model

To understand the ultrafast dynamics of the injection locking, we performed numerical simulations and theoretical analysis based on a simple equivalent circuit[36] ($LC$ oscillator) shown in Fig. 4a. Here, $v(t)$ is the oscillation voltage, $i(v)$ is the alternating current flowing through the RTD. For the current–voltage characteristics of the RTD, we assumed a cubic function, $i(v) = -\alpha v + \gamma v^3$, where $\alpha$ and $\gamma$ are positive constants (Fig. 4b). $G_L$ is the load conductance of the antenna that accounts for the radiation loss and is equal to $\alpha/2$ for the maximum radiation power[37]. The injection signal ($v_{inj}$) is modeled as a voltage source connected in series with the antenna conductance. The circuit equation becomes Van der Pol equation with external forcing term, which describes synchronization (injection locking) phenomena[38].

$$\ddot{\xi} + \epsilon\left(\xi^2 - 1\right)\dot{\xi} + \xi = -\epsilon\dot{\xi}_{inj} \qquad (1)$$

Here, $\xi = \sqrt{6\gamma/\alpha}\,v$ is the dimensionless oscillation amplitude, $\xi_{inj} = \sqrt{6\gamma/\alpha}\,v_{inj}$ is the dimensionless injection amplitude, $\tau = \omega_0 t$ ($\omega_0 = 1/\sqrt{LC}$) is the dimensionless time, and $\epsilon = \alpha\sqrt{L/C}/2$ is the nonlinearity parameter. The over-dot denotes the derivative with respect to $\tau$. As shown below, the time scale of the signal decay allows us to estimate the value of $\epsilon$ as around 0.01. This enables us to simulate

the RTD oscillator without assuming specific values for the circuit parameters ($C, L, \alpha,$ and $\gamma$). To simulate the typical situation with a short coherence time (Figs. 1 and 2), we only considered a single injection pulse. This is because the oscillation phase changes in a random fashion before the next injection pulse comes and cumulative effect does not exist. To mimic the phase fluctuation and data accumulation process in the experiment, we numerically solved Eq. (1) with twenty different initial phases and took the average of them ($\langle\xi\rangle$).

Figure 4d shows the simulated injection locking dynamics ($\langle\xi\rangle$, blue trace, left axis) induced by the injection pulse (orange trace, right axis). Although we use almost single-cycle, broadband THz pulses (Fig. 1c), only a small part of the spectrum should be injected into the RTD oscillator due to the frequency filtering effect of the antenna and $LC$ circuit. Therefore, we assumed multi-cycle injection waveform (see Methods for detail). The bottom axis is the dimensionless time $\tau$ and the top axis is the corresponding time when $\omega_0/2\pi = 340$ GHz. After the zero-signal due to the phase fluctuation, we see finite signal reflecting the phase-locking effect. The same waveform repeats at the rate of $f_{rep}$, which means that its frequency spectrum consists of comb modes at $nf_{rep}$ (i.e., injection locking to $nf_{rep}$). The build-up time of the phase-locked signal is around 50 ps, which reproduces the experimental results (Fig. 1f). In this model, the time scale of the injection locking is solely determined by the nonlinear parameter $\epsilon$. Therefore, it is advantageous to have large values of $\alpha$ and $L$, and a small value of $C$ for achieving a fast response. This provides a design guideline for the RTD device aimed at achieving a broad modulation bandwidth. After the signal reaches its maximum, the amplitude gradually decreases and becomes stable after ~250 ps, as observed in the experiment (Fig. 2a). The decay time here strongly depends on the value of $\epsilon$, which

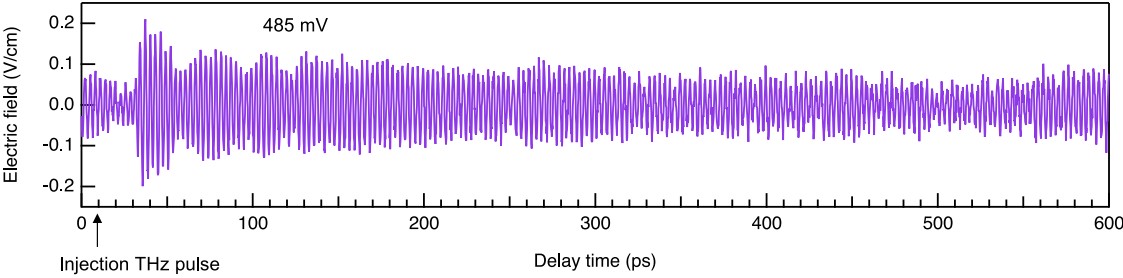

**Fig. 3 | RTD emission with long coherence time.** Oscillation signal is observed before the injection THz pulse arrived at 9 ps. The signal intensity is much stronger than those shown in Fig. 1 due to the cumulative effect by multiple injection THz pulses.

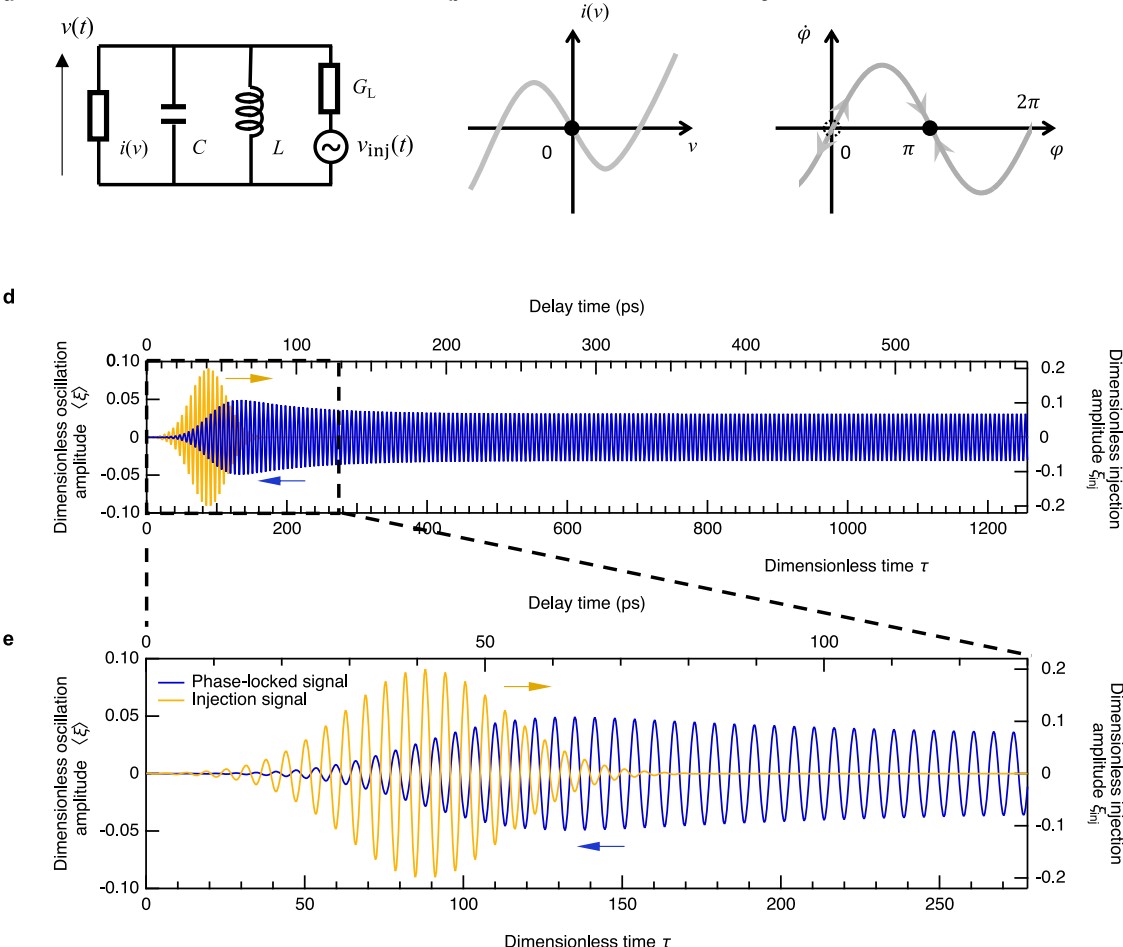

**Fig. 4 | Analysis based on Van der Pol model. a** Equivalent circuit. $G_L$: load conductance, $C$: capacitance, $L$: inductance, $i(v)$: RTD current, $v(t)$: oscillation voltage. **b**. Current–voltage characteristic approximated by a cubic function. We considered the simplest case of the bias voltage at the center of the NDC region[36] (inflection point), but similar results can be obtained at other bias voltages. **c** Phase response function of the forced Van der Pol oscillator, showing the stable steady-state solution at π. **d** Simulated waveform (blue, left axis) phase-locked by the injection signal (orange, right axis). **e** Magnified view of (**d**), showing the anti-phase locking behavior.

allowed us to estimate its value. Finally, if we look at the oscillation phase (Fig. 4e), this model reproduces the anti-phase locking behavior observed in the experiment (Fig. 2e).

The anti-phase locking nature can be explained by the following phase response function of the forced Van der Pol oscillator (see Methods for derivation):

$$\dot{\varphi} = \frac{\epsilon \xi_{\text{inj,o}}}{2\xi_0} \sin(\varphi) \quad (2)$$

Here $\xi_{\text{inj,o}} > 0$ and $\xi_0 > 0$ are the amplitudes of the injection signal and the oscillator, respectively. $\varphi(\tau)$ is the phase shift between the injection signal and the oscillator. Figure 4c shows the phase response function ($\dot{\varphi}$ vs $\varphi$). In the steady state ($\dot{\varphi} = 0$), there are two solutions at $\varphi = 0$ and π. However, at $\varphi = 0$, even a slight phase fluctuation toward the positive (negative) direction is amplified by positive (negative) $\dot{\varphi}$, indicating this is an unstable solution. In contrast, phase fluctuation around $\varphi = \pi$ is suppressed, making it the only stable solution, and consequently, the oscillator exhibits anti-phase locking behavior.

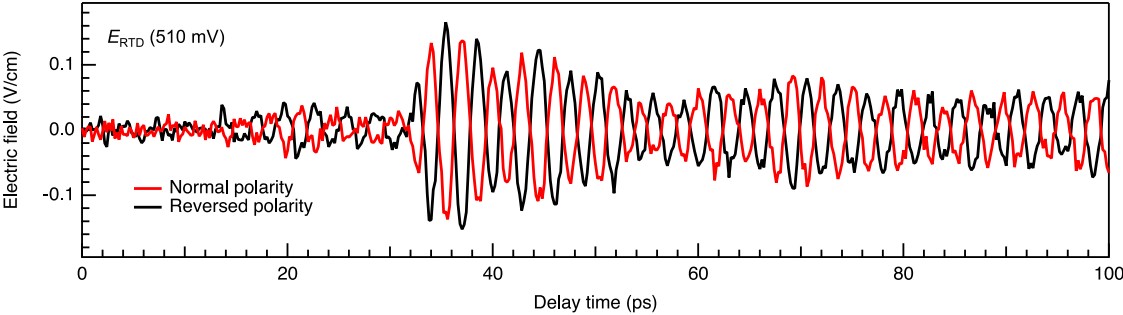

**Fig. 5 | Oscillation phase control.** The polarity of the injection THz pulse is reversed by flipping the sign of the voltage applied to the THz pulse emitter (photo-conductive antenna).

## Oscillation phase control

The forced Van der Pol model predicts a constant phase shift between the injection signal and the electronic oscillator. This allows us to arbitrarily control the oscillation phase by changing the phase of the injection signal. For a simple demonstration, we performed a similar experiment as Fig. 1d, but with polarity-reversed (π phase shifted) injection THz pulse. The red trace in Fig. 5 shows the same data as plotted in Fig. 1d, while the black trace shows the data taken with polarity-reversed injection THz pulse. As expected, the oscillation phase of the RTD emission is also reversed.

At this point, let us briefly mention the apparent delay in the RTD emission signal. In Fig. 5, it looks like the signal emerges around ~14 ps, whereas the injection THz pulse comes around ~9 ps. However, considering that the RTD emission signal gradually rises from zero, the signal immediately after the injection THz pulse should simply falls below the noise level. This inevitably produces certain time window where signal is not visible. At this stage, it is not feasible to experimentally ascertain whether there is a delay or not.

## Discussion

Phase control of THz electronic oscillators is an indispensable prerequisite for realizing many functionalities and applications such as beam forming/steering[39], radar and high-capacity wireless communications. Injection-locked RTD THz oscillators are promising devices for such operation because they allow phase control by electronic signal (bias voltage)[40]. This gives hope that high-speed phase modulation on the order of tens of GHz will be possible. However, the corresponding modulation period reaches as short as several tens of picoseconds, which is comparable to the time scale of the transient dynamics observed in this paper. This indicates that the maximum modulation bandwidth will be limited by the dynamical response time. This also shows the significance of understanding and controlling the ultrafast dynamics of the injection-locked THz oscillator. At the same time, ultrafast time-domain measurement of the THz electronic oscillators in the free-running state is also important to understand their phase fluctuation characteristics. For such purposes, single-shot THz TDS technique[41,42] that can capture non-repetitive waveform is necessary instead of the method presented in this paper.

It is worth comparing with similar optical sampling measurements performed in THz quantum cascade lasers (QCLs)[43,44]. In QCLs, offset-free THz pulses are injected and used as seed light for lasing via stimulated emission. Consequently, the emitted THz wave has the same phase as the THz pulse[45]. The in-phase emission is a hallmark of the laser oscillator. In contrast, we observed anti-phase emission in RTD, and revealed synchronization phenomena of limit cycle oscillators as a phase-locking mechanism. Although the experimental data (CW THz wave measured in the time domain) appears very similar to what is observed in QCLs, the underlying physical processes are totally different due to the distinct oscillation mechanisms between QCL (laser) and RTD (*LC* oscillator). Although THz QCLs and RTDs have much in common[46,47], our results show that the locking dynamics are clearly different and provide entirely new insights into THz electronic oscillators.

In conclusion, we showed that optical sampling technique is applicable to THz electronic oscillators by injection locking. This phase-resolved measurement allows us to directly monitor the ultrafast oscillation dynamics in response to external signal, which will be of significant importance for future applications such as ultrafast modulation in THz wireless communication. Based on the successful modeling using Van der Pol oscillator, we also demonstrated oscillation phase control through the phase of the injection waveform. This method will be universally applicable to other types of electronic limit cycle oscillators and provide a new tool for characterizing and controlling ultrafast response of THz electronic oscillators.

## Methods

### RTD device

We used an RTD oscillator made of AlAs/GaInAs double barrier structure made on 0.2-mm-thick InP substrate[17]. The current–voltage characteristic is shown in Supplementary Fig. S1a. The oscillation frequency ranges from 0.322 THz to 0.342 THz depending on the applied bias voltage. A typical spectrum, with a linewidth of ~3 MHz, measured by a conventional heterodyne down-conversion method is shown in Fig. S1b. The typical output power was around 10 µW. For the output power extraction, a bowtie antenna was integrated[33] and the RTD chip was mounted on a hyper hemispherical silicon lens with a diameter and thickness of 6 mm and 7.56 mm, respectively. To match the focal point of the silicon lens and the position of the RTD, we put silicon substrate with a thickness of 0.6 mm between the lens and RTD chip (Fig. 1b).

### Experimental setup

We used a mode-locked Ti:Sapphire laser (Spectra-Physics, Tsunami) that delivers 100 fs optical pulses (~800 nm center wavelength and ~850 mW average power) at 81 MHz repetition rate. Coherent THz pulses are generated by a large-area photo-conductive antenna[48]. For the electric field sampling, we employed electro-optic sampling method with a 2-mm-thick ZnTe crystal. A 2-mm-thick MgO crystal was attached on the ZnTe crystal. The delay time is limited to 600 ps simply by the length of the delay stage. See Supplementary Fig. S2a for the schematics.

### Data acquisition by double modulation

Taking two waveforms (with and without the bias voltage on RTD) of 600 ps takes a long time (~ 30 min each) and the long-term fluctuation during the measurements makes it difficult to obtain tiny RTD emission signal. To avoid this long-term fluctuation, we employed double modulation technique. We used a two-channel function generator to modulate the bias voltage for the photo-conductive antenna at 100 kHz with a 50 % duty cycle and the bias voltage for the RTD oscillator at the half frequency (50 kHz) with a 25 % duty cycle. See

Supplementary Fig. S2b for the timing chart. The electro-optic signal ($f(t)$) is periodic with the following unit signal:

$$E(t) = \begin{cases} E_{\text{on}} & (0 \le t \le T) \\ 0 & (T \le t \le 2T) \\ E_{\text{off}} & (2T \le t \le 3T) \\ 0 & (3T \le t \le 4T) \end{cases} \quad (2)$$

where $T = 5$ μs. The Fourier expansion of this signal becomes as follows:

$$E(t) = \frac{E_{\text{off}} + E_{\text{on}}}{4} + \sqrt{2}\frac{E_{\text{on}} - E_{\text{off}}}{\pi}\sin\left(2\pi f t + \frac{\pi}{4}\right) \\ + \frac{E_{\text{off}} + E_{\text{on}}}{\pi}\sin(4\pi f t) + \cdots \quad (3)$$

Here $f = 1/(4T)$. By using the $f$ and $2f$ components ($E_f$ and $E_{2f}$, respectively) measured by a dual frequency lock-in amplifier, we obtained $E_{\text{off}}$ and $E_{\text{RTD}} = E_{\text{on}} - E_{\text{off}}$ as follows:

$$E_{\text{off}} = \frac{\pi}{4}\left(2E_{2f} - \sqrt{2}E_f\right) \quad (4)$$

$$E_{\text{RTD}} = \frac{\pi}{\sqrt{2}}E_f \quad (5)$$

We have checked this modulation scheme works correctly by confirming that $E_{\text{off}}$ and $E_{\text{RTD}}$ obtained through Eqs. (4) and (5) agree with the ones obtained from two different waveform measurements with and without the bias voltage on RTD.

### Injection waveform for numerical simulation

We assumed multi-cycle injection waveform represented by the following function with the Gaussian envelope and carrier frequency of $\nu_0$ (free-running frequency of the oscillator):

$$\xi_{\text{inj}} = \xi_{\text{inj}}^0 e^{-\frac{(\tau - \tau_0)^2}{2\sigma^2}}\cos(2\pi\nu_0(\tau - \tau_0)) \quad (6)$$

As reasonable parameters that reproduce the experimental results, we used $\xi_{\text{inj}}^0 = 0.2$, $\tau_0 = 28\pi$ and $\sigma = 8\pi$. The ratio of the full-width at half maximum to the center frequency of the power spectrum is ~10, which is close to the quality factor of 4 reported for a similar RTD device[32]. Deriving the injection amplitude $\xi_{\text{inj}}^0$ is not straightforward due to several uncertainties such as coupling efficiency and possible electric field enhancement by the antenna electrodes. However, the injection power dependence allows us to determine the appropriate range for $\xi_{\text{inj}}^0$ as follows. We found that the simulation results show almost the same dynamics as long as the injection amplitude $\xi_{\text{inj}}^0$ is far below two (the free-running amplitude of the limit cycle oscillator). In this regime, only the phase-locked signal amplitude, $\langle\xi\rangle$ changes in proportion to the injection amplitude $\xi_{\text{inj}}^0$, which is actually the case in our experiment. Therefore, we performed simulations in this weak injection regime ($\xi_{\text{inj}}^0 = 0.2$ specifically).

### Phase response function of forced Van der Pol oscillator

The anti-phase locking nature of the forced Van der Pol oscillator can be understood based on its phase response function[49,50]. We calculate the phase response function under the following injection signal:

$$\xi_{\text{inj}} = \xi_{\text{inj,o}}\sin\left(\Omega_{\text{inj}}\tau\right) \quad (7)$$

Here $\xi_{\text{inj,o}} > 0$ is the amplitude and $\Omega_{\text{inj}}$ is the frequency which is very close to the free-running one, i.e., $\Omega_{\text{inj}} \sim 1$. As a first approximation, we assume that the oscillator is injection-locked to $\Omega_{\text{inj}}$ as follows:

$$\xi = \xi_0\sin\left(\Omega_{\text{inj}}\tau + \varphi(\tau)\right) \quad (8)$$

Here $\xi_0 > 0$ is the amplitude and $\varphi(\tau)$ is the phase shift. We consider a weak injection case where the amplitude $\xi_0$ does not change and only the phase (or equivalently, frequency) changes by the injection. By substituting Eqs. (7) and (8) into (1) and equating the coefficients of $\sin(\Omega_{\text{inj}}\tau + \varphi(\tau))$ and $\cos(\Omega_{\text{inj}}\tau + \varphi(\tau))$ terms separately to zero, we obtain the following equations:

$$-\xi_0\left(\Omega_{\text{inj}} + \dot{\varphi}\right)^2 + \xi_0 = -\epsilon\Omega_{\text{inj}}\xi_{\text{inj,0}}\sin(\varphi) \quad (9)$$

$$\xi_0\ddot{\varphi} + \epsilon\xi_0\left(\Omega_{\text{inj}} + \dot{\varphi}\right)\left(\frac{\xi_0^2}{4} - 1\right) = -\epsilon\Omega_{\text{inj}}\xi_{\text{inj,0}}\cos(\varphi) \quad (10)$$

Here we ignored the third-harmonic term since the first approximation is concerned only with the fundamental frequency[38]. When the phase $\varphi(\tau)$ varies slowly during one period of oscillation ($2\pi/\Omega_{\text{inj}}$) due to the weak injection, we have $1 \gg \dot{\varphi} \gg \ddot{\varphi}$. In this condition, we have the following phase response function from Eq. (9)

$$\dot{\varphi} = \frac{1 - \Omega_{\text{inj}}^2}{2\Omega_{\text{inj}}} + \frac{\epsilon\xi_{\text{inj,0}}}{2\xi_0}\sin(\varphi) \\ \approx \frac{\epsilon\xi_{\text{inj,0}}}{2\xi_0}\sin(\varphi) \quad (11)$$

where we neglected the first term in the right-hand side since $\Omega_{\text{inj}} \sim 1$. In the steady state ($\dot{\varphi} = 0$), we have only one stable solution at $\varphi = \pi$, which means the oscillator shows anti-phase locking behavior. In this condition, we have the following relation from Eq. (10):

$$\xi_0\left(\frac{\xi_0^2}{4} - 1\right) - \xi_{\text{inj,0}} = 0 \quad (12)$$

Since we deal with the weak injection case ($\xi_0 \gg \xi_{\text{inj,0}}$), we have $\xi_0 \approx 2$. This means that the oscillation amplitude does not change from the free-running state, which is consistent with the initial assumption.

## Data availability

All the raw and processed data used in the figures in the main text and Supplementary Information are available in Zenodo repository (https://doi.org/10.5281/zenodo.10910331)[51].

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

## Acknowledgements

The authors thank M. Asada and S. Suzuki for valuable discussions. We also thank Y. Nishida for supporting device preparation. This work was supported in-part by ROHM joint research program (T.A.), JST PRESTO (Grant No. JPMJPR21B1, T.A.), JST CREST (Grant No.

JPMJCR1534 and JPMJCR21C4, M.F.), and JST ACCEL (Grant No. JPMJMI17F2, K.T.).

## Author contributions
T.A. conceived and conducted the experiment. J.K. and T.M. fabricated the RTD chip. N.N., M.F., and T.N. designed and prepared the RTD device. T.A. performed numerical simulation with inputs from K.T. T.A. and M. F. discussed on the analysis of data with inputs from T.N. and K.T. K.T. supervised the project. T.A. wrote the manuscript with contributions from all authors.

## Competing interests
The authors declare no competing interests.
