## [Peer Review File · Nature Communications]

Phase-resolved measurement and control of ultrafast dynamics in terahertz electronic oscillatorsREVIEWER COMMENTS

Reviewer #1 (Remarks to the Author):

Dear authors, thank you for submitting a very interesting paper.

The investigation of injection locking behavior of electronic THz oscillators is of very high interest for the future development of electronic THz systems. The identification of anti-phase locking and the temporal analysis of the injection locking process are crucial for the application of resonant tunneling diodes, or any other injection-locked electronic oscillator source, in future THz wireless data transmission systems.

The measured data support the assumed van der Pol model of the RTD oscillator circuit. The applied filtering and exclusion of echos not related to the DUT appear justified, applying established THz measurement techniques. The quality of the obtained data is very high, owing to the applied double-frequency lock-in measurement technique.

The discussion of the anti-phase locking behavior could be improved by explicitly showing the stability condition used in lines 210-216 and 447-450. (Why is $\phi = 0$ not a stable solution?)

Furthermore, it is not immediately clear from which data set the phase shift in Fig. 2e was derived. The anti-phase locking behavior is an important finding in the paper.

Regarding the possible modulation bandwidth of injection-locked RTD, it would be interesting to put the observed injection locking time scale of 35-50 psec (lines 119-129) into perspective with the RTD's resonant circuit parameters L and C, and the RTD's IV parameters alpha and gamma (lines 170-190).

Overall, the methodology used in the presented work is sound, and the work meets the highest standards in the field. Given the clear and detailed description of the experiment and of the applied mathematical principles, also in the supplemental material, it should be quite possible to reproduce the presented work in a well equipped laboratory.

The reviewer suggests that the paper be published after addressing above comments.

Reviewer #2 (Remarks to the Author):

The manuscript reports on the injection seeding of a resonant tunneling diode in the THz region. The upper oscillation frequencies of RTD can now reach almost 2 THz, which is very important and interesting. However, the THz detection techniques used for characterizing were conventional GHz detectors. In the THz range time-resolved techniques derived from fs-laser are available. These have been applied in the present work to lock and time-resolve the emission of RTDs. This in principle is interesting. However, only THz oscillations with oscillation frequencies corresponding to that of the RTD are shown. The observed time-traces are unclear and the model to explain them is not convincing. Therefore, I cannot recommend publication in Nature Comms.

Here are more detailed critical points.

- 1) In Fig. 1 the injection THz pulse is not visible? In Fig.1 c) a reference pulse is shown with a much higher amplitude. Where is the injection pulse in d) and e).
- 2) The authors claim a relative amplitude of larger 1. How was this relation obtained?
- 3) Electro optic sampling allows to calibrate the THz field amplitude. Why did the authors not give absolute values of the THz electric field?
- 4) The observed time trace could also result from a strong absorption of a cavity. The sign reversal of the emission points even more to this – since it is a linear process.

- 5) The employed model uses a harmonic drive which is certainly not the case – c.f. Fig. 1c it is a half cycle pulse with a broad spectrum. Therefore, the model cannot be used to explain the observed data
- 6) The observed delay points rather to a signal emitted from a position further away
- 6) Complementary measurements with c.w. detectors are required. The emission intensity at the c.w. detectors should show a strong effect for successful injection locking. Also complementary linewidth measurements with RF techniques are required.

Reviewer #3 (Remarks to the Author):

The title of the manuscript by T. Arikawa that I reviewed is “Phase-resolved measurement and control of ultrafast dynamics in terahertz electronic oscillator”.

This article described and discussed the phase-resolved measurement that the optical sampling technique is applicable to THz electronic oscillators by injection locking. The paper exhibits a commendable level of proficiency in terms of its structural organization, discussion, and other relevant aspects. Furthermore, the experimental findings presented within are aesthetically pleasing and demonstrate a high degree of coherence and reliability.

Undoubtedly, the achievement of anti-phase locking represents a highly notable breakthrough. Nevertheless, as stated by the author, there exist prior experimental studies employing QCLs, yet regrettably, the underlying physical principle is not novel.

The utilization of QCL in previous reports has resulted in a dearth of originality concerning proof-of-principle, mechanism elucidation, and experimental procedures. Consequently, achieving publication in a journal such as Nature Communications, which places significant emphasis on general impact, would pose challenges at this particular stage.

The authors' argument would be further reinforced by incorporating an actual demonstration of the application to THz wireless communications. This optical sampling method exhibits remarkable qualities, making it a commendable approach, so I suggest that the present version be submitted to specialized academic journals.

The authors' efforts are commendable; however, certain modifications are necessary to meet the publication standards of Nature Communications.

We greatly appreciate the reviewers' careful reading and insightful comments, which are very helpful for improving the manuscript. We have implemented their comments and suggestions. Below, we will respond to each of their comments. In the revised manuscript, revised and newly added texts are highlighted in blue and deleted texts are struck through.

Response to reviewer 1

Comment (1)-0

“Dear authors, thank you for submitting a very interesting paper.

The investigation of injection locking behavior of electronic THz oscillators is of very high interest for the future development of electronic THz systems. The identification of anti-phase locking and the temporal analysis of the injection locking process are crucial for the application of resonant tunneling diodes, or any other injection-locked electronic oscillator source, in future THz wireless data transmission systems.

The measured data support the assumed van der Pol model of the RTD oscillator circuit. The applied filtering and exclusion of echos not related to the DUT appear justified, applying established THz measurement techniques. The quality of the obtained data is very high, owing to the applied double-frequency lock-in measurement technique.”

Response to comment (1)-0

Thank you for acknowledging the importance of our paper. We appreciate your positive feedback.

Comment (1)-1

“The discussion of the anti-phase locking behavior could be improved by explicitly showing the stability condition used in lines 210-216 and 447-450. (Why is $\phi = 0$ not a stable solution?)”

Response to comment (1)-1

We thank the reviewer for this suggestion. In the revised manuscript, we added the following sentences in lines 230-235 to explicitly show the stability condition. “Figure 4c shows the phase response function ($\dot{\phi}$ vs ϕ). In the steady state ($\dot{\phi} = 0$), there are two solutions at $\phi = 0$ and π . However, at $\phi = 0$, even a slight phase fluctuation

toward the positive (negative) direction is amplified by positive (negative) ϕ , indicating this is an unstable solution. In contrast, phase fluctuation around $\varphi = \pi$ is suppressed, making it the only stable solution, and consequently, the oscillator exhibits anti-phase locking behaviour.”

Comment (1)-2

“Furthermore, it is not immediately clear from which data set the phase shift in Fig. 2e was derived. The anti-phase locking behavior is an important finding in the paper.”

Response to comment (1)-2

We thank the reviewer for pointing this out. To determine the phase shift at each bias voltage, we applied the same procedure described in the previous subsection (lines 135-141) to all the data set shown in Fig. S3 (bias voltage dependence of the time-domain signal). To explicitly show this point, we modified the explanation as follows (lines 155-161). “Finally, we determined the phase of the RTD oscillation at each bias voltage by applying the same fitting procedure described in the previous subsection. The phase of the injection THz wave at the RTD oscillation frequency can be obtained from the inset of Fig. 1c. By subtracting these two, the bias voltage dependence of the phase shift was determined, and the results are summarized in Fig. 2e. Although it shows some deviation, again below 480 mV, the data points are concentrated around π (i.e., anti-phase locking) regardless of the bias voltage.”

Also, in order to underscore the significance of the phase shift, we introduced a dedicated subsection with the title "Anti-phase Locking." in line 135. The caption of Fig. 2e is also modified as follows. “e. Phase shift determined from the oscillation phase of the time-domain waveform at each bias voltage.”

Comment (1)-3

“Regarding the possible modulation bandwidth of injection-locked RTD, it would be interesting to put the observed injection locking time scale of 35-50 psec (lines 119-129) into perspective with the RTD's resonant circuit parameters L and C, and the RTD's IV parameters alpha and gamma (lines 170-190).”

Response to comment (1)-3

We appreciate the reviewer's valuable suggestion. In our model, the time scale of the

injection locking is exclusively governed by the nonlinear parameter $\epsilon = \alpha\sqrt{L/C}/2$. To attain a high modulation frequency, it is advantageous to have large values of α and L , and a small value of C . This offers a design guideline for the RTD device aimed at achieving a broad modulation bandwidth. We added the following sentences in lines 213-216. “In this model, the time scale of the injection locking is solely determined by the nonlinear parameter ϵ . Therefore, it is advantageous to have large values of α and L , and a small value of C for achieving a fast response. This provides a design guideline for the RTD device aimed at achieving a broad modulation bandwidth.”

At the same time, we omitted the following sentence in lines 216-217: "This result shows that the build-up time is determined by the duration of the injection signal." This decision was made considering that, even with the same injection signal, the rate of signal increase varies depending on the value of ϵ .

Comment (1)-4

“Overall, the methodology used in the presented work is sound, and the work meets the highest standards in the field. Given the clear and detailed description of the experiment and of the applied mathematical principles, also in the supplemental material, it should be quite possible to reproduce the presented work in a well equipped laboratory.

The reviewer suggests that the paper be published after addressing above comments.”

Response to comment (1)-4

We appreciate the reviewer's positive evaluation of our work. We believe we have thoroughly addressed the concerns raised by the reviewer.

Response to reviewer 2

Comment (2)-0

“The manuscript reports on the injection seeding of a resonant tunneling diode in the THz region. The upper oscillation frequencies of RTD can now reach almost 2 THz, which is very important and interesting. However, the THz detection techniques used for characterizing were conventional GHz detectors. In the THz range time-resolved techniques derived from fs-laser are available. These have been applied in the present work to lock and time-resolve the emission of RTDs. This in principle is interesting. However, only THz oscillations with oscillation frequencies corresponding to that of the RTD are shown. The observed time-traces are unclear and the model to explain them is not convincing. Therefore, I cannot recommend publication in Nature Comms. Here are more detailed critical points.”

Response to comment (2)-0

Thank you for your invaluable feedback on our paper. We understand your concerns about the clarity of the data and the model used to support our claim that the observed time-traces indeed represent THz radiation emitted from RTD. However, we believe that the source of your concerns lies in the unclear description in the manuscript, leading to some misunderstandings. In fact, some of the issues raised may not be accurate reflections of the content. To enhance the manuscript's clarity, we have diligently revised it as outlined below. We are confident that these revisions will address all concerns.

Comment (2)-1

“In Fig. 1 the injection THz pulse is not visible? In Fig.1 c) a reference pulse is shown with a much higher amplitude. Where is the injection pulse in d) and e).”

Response to comment (2)-1

Figures 1d and 1c show the waveform difference between bias voltage on and off. As a result, the injection THz pulse is canceled out. This is why the injection THz pulse is not visible. Although this point is mentioned in lines 92-96, we have added the following sentence to the figure caption of Fig.1d to avoid confusion: “Injection THz pulse is nearly perfectly canceled by the double modulation technique.”

Comment (2)-2

“The authors claim a relative amplitude of larger 1. How was this relation obtained?”

Response to comment (2)-2

The relative amplitude presented in Fig. 1g was obtained from the experimental results: $|\tilde{E}_{\text{on}}|/|\tilde{E}_{\text{off}}|$, where \tilde{E}_{on} and \tilde{E}_{off} represent the Fourier transforms of the waveforms with bias voltage on and off, respectively. A relative amplitude larger than 1 at the RTD emission frequency indicates the presence of THz wave emitted from the RTD in addition to the incident THz wave (\tilde{E}_{off}). This is the reason why we can eliminate the possibility of absorption. In cases of absorption, as pointed out by the reviewer in comment (2)-4 below, a similar time trace can also occur. However, in such instances, the phase of the oscillation waveform is shifted by π from the incident wave, leading to destructive interference and a relative amplitude less than 1. In the present case, even though the RTD emission is also π phase shifted (anti-phase locking), the amplitude of the RTD emission exceeds that of the injection THz wave at the specific frequency, resulting in a relative amplitude greater than 1.

Comment (2)-3

“Electro optic sampling allows to calibrate the THz field amplitude. Why did the authors not give absolute values of the THz electric field?”

Response to comment (2)-3

We appreciate the reviewer for bringing this to our attention. As we have a record of the conversion factor from the electro-optic signal to the electric field value, we changed the vertical axis of the waveform to electric field.

Comment (2)-4

“The observed time trace could also result from a strong absorption of a cavity. The sign reversal of the emission points even more to this – since it is a linear process.”

Response to comment (2)-4

As we explained in the response to comment (2)-2 above, we can rule out the possibility of absorption as the cause for the experimental results obtained in this study. To clearly show this point, we modified the sentences in lines 103-109 as follows. “However, a

similar wave form can also arise due to absorption. To ascertain the cause of the signal, we calculated the Fourier transform (\tilde{E}_{on} and \tilde{E}_{off}) and checked the amplitude ratio ($|\tilde{E}_{\text{on}}|/|\tilde{E}_{\text{off}}|$). As shown by the red curve in Fig. 1g, a sharp peak above one at 0.340 THz is seen, indicating the presence of THz wave emitted from the RTD in addition to the incident THz wave. This allows us to eliminate the possibility of absorption as the cause for the oscillation observed in Fig. 1d.”

Comment (2)-5

“The employed model uses a harmonic drive which is certainly not the case – c.f. Fig. 1c it is a half cycle pulse with a broad spectrum. Therefore, the model cannot be used to explain the observed data”

Response to comment (2)-5

With all respect to the reviewer, it appears there might have been a misunderstanding or oversight in the explanation provided. *Although we inject broadband THz pulse, only a limited portion of the spectral component resonant to the antenna and the RTD LC circuit is received.* This frequency filtering effect has been carefully addressed in the text and Methods section (“Injection waveform for numerical simulation” in lines 440-457). However, to enhance the manuscript's clarity, we revised the sentences in lines 204-208 as follows: “Although we use almost single-cycle, broadband THz pulses (Fig. 1c), only a small part of the spectrum should be injected to the RTD oscillator due to the frequency filtering effect of the antenna and LC circuit. Therefore, we assumed multi-cycle injection waveform (see Methods for detail).”

Comment (2)-6

“The observed delay points rather to a signal emitted from a position further away”

Response to comment (2)-6

Thank you for pointing this out. At first glance, it may seem that the signal emitted from the RTD is occurring with a delay compared to the incident THz pulse. For example, in Fig. 5, it looks that the signal emerges around ~14 ps, whereas the injection THz pulse comes around ~9 ps. However, considering that the RTD emission signal gradually rises from zero, the signal immediately after the injection THz pulse should simply falls below the noise level. This inevitably produces certain time window where signal is not visible.

At this stage, it is not feasible to experimentally ascertain whether there is a delay or not. To address this concern, we added the following explanation in lines 245-250. “At this point, let us briefly mention the apparent delay in the RTD emission signal. In Fig. 5, it looks that the signal emerges around ~ 14 ps, whereas the injection THz pulse comes around ~ 9 ps. However, considering that the RTD emission signal gradually rises from zero, the signal immediately after the injection THz pulse should simply falls below the noise level. This inevitably produces certain time window where signal is not visible. At this stage, it is not feasible to experimentally ascertain whether there is a delay or not.”

Comment (2)-7

“Complementary measurements with c.w. detectors are required. The emission intensity at the c.w. detectors should show a strong effect for successful injection locking. Also complementary linewidth measurements with RF techniques are required.”

Response to comment (2)-7

Thank you for highlighting the importance of basic characterization. The emission intensity, measured by a conventional square law detector, is indeed depicted in Fig. 2d as the blue trace. To explicitly convey this information, we revised the sentence in lines 150-152 as follows: “As plotted in Fig. 2d by the red circles, the overall trend is similar to the output intensity in the free-running state, measured by a conventional square law detector (blue), especially above 480 mV.” The output power changes depending on the bias voltage. As pointed out by the reviewer, a strong oscillator (with high output power) is challenging to injection lock. However, as demonstrated in Fig. S3, we were able to injection lock the RTD oscillator at all examined bias points. This is likely because the power of the injection THz pulse significantly exceeds the threshold power required for injection locking, and the RTD output power variation has minimal impact. The result of the spectrum measurement is added as Fig. S1b, and the linewidth value (~ 3 MHz) is mentioned in the “RTD device” in Methods section as follows (lines 402-403). “A typical spectrum, with a linewidth of ~ 3 MHz, measured by a conventional heterodyne down-conversion method is shown in Fig. S1b.”

Response to reviewer 3

Comment (3)-0

“The title of the manuscript by T. Arikawa that I reviewed is “Phase-resolved measurement and control of ultrafast dynamics in terahertz electronic oscillator”.

This article described and discussed the phase-resolved measurement that the optical sampling technique is applicable to THz electronic oscillators by injection locking. The paper exhibits a commendable level of proficiency in terms of its structural organization, discussion, and other relevant aspects. Furthermore, the experimental findings presented within are aesthetically pleasing and demonstrate a high degree of coherence and reliability.”

Response to comment (3)-0

We are very pleased to receive such high praise for our research.

Comment (3)-1

“Undoubtedly, the achievement of anti-phase locking represents a highly notable breakthrough. Nevertheless, as stated by the author, there exist prior experimental studies employing QCLs, yet regrettably, the underlying physical principle is not novel.

The utilization of QCL in previous reports has resulted in a dearth of originality concerning proof-of-principle, mechanism elucidation, and experimental procedures. Consequently, achieving publication in a journal such as Nature Communications, which places significant emphasis on general impact, would pose challenges at this particular stage.”

Response to comment (3)-1

Thank you for the comment on the relation to the prior research on QCL. As the reviewer noted, the experimental procedures are indeed similar, and the experimental data (CW THz wave measured in the time domain) appears very much alike. However, let us emphasize that the underlying physical processes are totally different due to the distinct oscillation mechanisms between QCL (laser) and RTD (LC oscillator). As opposed to the stimulated (in-phase) emission in QCL, we revealed synchronization phenomena of limit cycle oscillators as a phase locking mechanism in RTD. *Our work with RTD has provided entirely new insights into THz electronic oscillators that were never anticipated from what is known in QCL.* Therefore, in our opinion, the criticism regarding novelty and

originality does not apply in this context. Considering the anticipated importance of room temperature THz electronic oscillators for various applications, our study holds significant general impact. To emphasize these points, we modified the sentences in lines 270-278 as follows. “In contrast, we observed anti-phase emission in RTD, and revealed synchronization phenomena of limit cycle oscillators as a phase locking mechanism. Although the experimental data (CW THz wave measured in the time domain) appears very similar to what is observed in QCLs, the underlying physical processes are totally different due to the distinct oscillation mechanisms between QCL (laser) and RTD (LC oscillator). Although THz QCLs and RTDs have much in common, our results show that the locking dynamics is clearly different and provide entirely new insights into THz electronic oscillators.”

Comment (3)-2

“The authors' argument would be further reinforced by incorporating an actual demonstration of the application to THz wireless communications.”

Response to comment (3)-2

Thank you for the comment on the demonstration. The current study offers a design guideline for THz electronic oscillators to achieve fast phase response (cf. lines 213-216 in the revised manuscript). This guideline is valuable for enhancing the modulation bandwidth and data rate of THz wireless communications based on phase modulation. THz wireless communication utilizing the phase modulation scheme has been successfully implemented with CMOS or optoelectronic technologies. However, when it comes to fundamental THz oscillators like RTD, the utilization of phase in wireless communication is still in the early stages of development. So far, injection locking phenomena has been utilized on the receiver side of wireless communications to achieve coherent detection, resulting in a sensitivity enhancement of 20 dB (ref. 33). Nevertheless, the modulation scheme used in this experiment was amplitude-shift keying. To implement the phase modulation scheme, achieving arbitrary phase control of the THz transmitter is essential. However, at this stage, only one proof-of-concept experiment of the phase control using RTD has been reported (ref 42). Our method, employing coherent THz pulses, can control the phase, but it is not suitable for serving as a phase modulator. This is because pulse-to-pulse arbitrary phase control of the THz pulses, which is necessary to convey information, is not feasible even with state-of-the-art techniques. *Therefore, the demonstration of phase modulation wireless communication using the THz electronic*

oscillator itself is a significant research topic that demands considerable time and effort. Due to the complexity and depth of this subject, we believe it is beyond the scope of the current paper.

Comment (3)-3

“This optical sampling method exhibits remarkable qualities, making it a commendable approach, so I suggest that the present version be submitted to specialized academic journals. The authors' efforts are commendable; however, certain modifications are necessary to meet the publication standards of Nature Communications.”

Response to comment (3)-3

In response to the reviewer's comments, we have emphasized the novelty of the results obtained with RTD. With these modifications, we believe that the significance of this study has become clear and aligns with the publication standards of Nature Communications.

REVIEWERS' COMMENTS

Reviewer #1 (Remarks to the Author):

Dear authors,
thank you for answering the questions and providing additional information.

Reviewer #2 (Remarks to the Author):

The authors have improved the manuscript. Their arguments are now much clearer and supported by the model and data. The manuscript is now suitable for publication

Reviewer #3 (Remarks to the Author):

The authors responded sincerely to the reviewers' comments.
In particular, the differences with respect to the rocking dynamics became clearer to the reader following the modification of the text, compared to the results of similar experiments using QCLs. The same applies to the description of THz wireless communications.

The Review respects the authors' effort and considers it worthy of publication in Nature Communications with this revision.